# Effects of urbanization and land expropriation on the livelihoods of peri-urban farmers in North Wollo Zone, Ethiopia

**Mesele Belay Zegeye** [ID]°*, **Moges Asmare Sisay**°, **Dagnaw Beza Ayalew**°

Department of Economics, Woldia University, Woldia, Ethiopia

° These authors contributed equally to this work.
* mesele99belay@gmail.com

## Abstract

The aim of this study was to examine the effects of urbanization and land expropriation on the livelihoods of Peri-urban farmers in North Wollo Zone, Ethiopia. Data were collected from 378 peri-urban farmers using a structured questionnaire and multistage sampling. Qualitative data were obtained through focus group discussions and key informant interviews. A binary logistic regression model was used to identify factors influencing livelihood diversification among expropriated farm households. The results revealed a significant trend of expropriation without fair compensation, stemming from the disregard of legal procedures of expropriation and compensation in the study area. On average, 33.06% of respondents were fully expropriated and evicted, while the majority (66.94%) lost only part of their agricultural land. Expropriated farmers faced numerous challenges during their livelihood diversification, including high inflation, lack of urban living skills, informal land sales at low prices, misuse of compensation, displacement, family conflicts, social disruption, and skill gaps. They also blamed that they did not get any government support in properly utilizing compensation money for livelihood diversification. The Logit model results showed that livelihood diversification is positively influenced by factors such as the household head's education, marital status, access to alternative land, fair compensation, off-farm employment, training, social networks, and livestock assets. It is negatively affected by farming as the primary occupation, household size, and expropriated land size. The study recommends that the government should help families of expropriated household's secure sustainable livelihoods through fair compensation and proper support.

## 1. Introduction

Urbanization is an occurrence with wavering peculiarities in different regions of the world. The world urban population has enlarged from 29.6% in 1950 to 53.6% in

**Data availability statement:** The data that support the findings of this study is openly available in the link https://zenodo.org/records/14616162.

**Funding:** The author(s) received specific funding for this work from Woldia University, Ethiopia.

**Competing interests:** The authors have declared that no competing interests exist.

2014 and is expected to range 66.4% by 2050. Urban expansion effects in land use change of nearby rural areas to meet the needs of urban areas via expropriation. Land use change in turn has an effect on the livelihoods of farming communities. To make urban expansion organized and sustainable, development policies need to be all-inclusive; the expropriation process should be apparent and participatory and livelihood restoration plan should be part of the urban expansion pattern [1,2] and [3,4]. Ethiopian economy has been among one of the fast growing economies in Africa and in the world at large. Between 2004 and 2019, the country is the only economy in the world to achieve over 10 percent growth [5]. Over the 2000s the main driver of the economic growth was an increase in productivity, which was accompanied by high rate of capital accumulation. Mainly the surge in public infrastructure spending was seen as a catalyst for the high growth rate. During the 2010s the main sources for growth were service and the boom in the construction. These sources on the other hand were directly related to urbanization [5] and [3,4].

The urbanization rate in Ethiopia is high and the main cause is not rural urban migration rather it is mainly through the internal expansion of cities, one of the sources being the reclassification of rural areas [5]. The country is expected to be among the ten countries with high urbanization rate in the period 2018–2050 [2]. This process needs the expansion of the urban areas, cities and towns in to the nearby rural areas. And it follows with high demand for basic services like housing, water, sanitation and energy in the urban areas.

Most of the time the urbanization process in developing countries follows a horizontal expansion, which is the same in Ethiopia. In order to fulfil the high demand for land in urban areas, the federal government had enacted different laws that enable local governments to expropriate land from farmers in the Peri-urban areas for public purposes. The regulation asserts that farmers, who are evicted from their lands', are entitled to land compensation in terms of cash [3,4,6,7]. The land expropriation and compensation process and amount were one of the sources of dissatisfaction among evicted farmers and there were several improvements in legalisation and the calculation of the compensation [8]. One of the areas where there is a high level of expansion of cities is North Wollo Zone of the Amhara national regional state, Ethiopia.

One of the cities in North Wollo zone where there is a high rate of expansion and reclassification of the rural areas as urban is Woldia, its capital city. The city has become a center of attraction for industrial and other investments. The other cities with high urbanization rate include Kobo, Mersa and Lalibela towns. These cities were showing a fast urbanization both as an investment and a residential center. The proximity to the capital and the good investment climate that exits in the area had make the town the choice of many investors. The high rate of urbanization is also creating a huge demand for resources and infrastructures [9].

This surge in investments demands high level of infrastructure and resource. The infrastructure demand includes housing, water and sanitation, energy, transport and different amenities. The most important resource in a highly urbanizing area is believed to be land. Similarly land in cities of North Wollo zone is the most sighted resource. This is because land is necessary for investment, housing and

infrastructure projects. In order to make avail these land the city is expanding towards the rural villages in the nearby, which is made through land expropriation and land compensation, reclassification the villages as urban areas. In addition the expansion of the urbanization also provides an opportunity for the farmers including employment, access to services and high income [4,8,9].

Land is one of the most precious resources in agrarian economies. It serves as a source of income for farming households, as a form of asset to accumulate wealth and it is transferred to generations [3,5,10]. The high rate of urbanization and the expansion of industries and investments in many countries, Ethiopia included, making land available for demanders has become a complicated process for governments. This is because in order to get the land, the governments have to expropriate land from the framers in peri-urban areas. Land expropriation is the process of compulsory taking of land by the government for public purposes like infrastructure, urban housing or investment projects in advance payment of compensation [11]. There were several studies from different perspectives on the process of land expropriation and land compensation worldwide and in Ethiopia.

These studies can be grouped in to three; studies on the legal process and legality of land expropriation, studies on the consequence of the land expropriation and studies on the livelihood diversification strategies of evicted farmer's or peri-urban farmers. Especially the expropriation and its consequences have been the subject of several studies. Land expropriation threatens social order [12] and also was found to be harmful to the health and subjective well-being [13,14]. This mainly comes from the income lost and the psychological effects of land expropriation [15]. On the other hand land expropriation also found to be raising farmers' value expectation of land [14].

The economic consequence of land expropriation is the other important topic which is extensively covered in the literature. Land expropriation and the farmer's entrepreneurial action were studied by [16] in Zhejiang province, China and they have found that the main important factor determining the entrepreneurial action of farmers' was land location. In addition the compensation size affects the farmer's entrepreneurial action negatively. A study by [17] compares the employment status of farmers before and after the land expropriation of land lost farmers and found that many farmers have changed their original employment, which was agriculture, and mainly either become migrant labourer or unemployed. The welfare effect, by using income and the level of individual happiness as proxy, of land expropriation was studied by [13] and they have found that there was an increase in income while there was a decline in the individual happiness. On the other hand a similar study in Vietnam has found that the welfare of land lost farmers has decreased mainly due to a reduction in the agricultural income and an inability to participate in the non-agricultural labour market [18]. In addition to these studies there are substantial studies that focus on the livelihood diversification strategies of farmers in peri-urban areas.

[10] has studied the impact of land expropriation on the livelihood of expropriated peri-urban farmers around Debre Markos city, Ethiopia. Their study has found that there were problems that the farmers faced which were related to livelihood including food insecurity, social and family disintegration. These were mainly due to the process of expropriation without fair and appropriate compensation in the city and it was reported as a main source of dissatisfaction. On the other hand a study by [19] has found that the livelihood of peri-urban farmers, who were evicted from their land, is negatively affected by the process of land expropriation and compensation. This is a cumulative result of the poor saving habits, lack of capacity to competition among the evicted farmers and the policy and institutional limitations of the government. In order to avoid these negative livelihood effects the evicted farmers tried to diversify their livelihood. A study by [20] and [19] has found that since there is no process of rehabilitation of evicted farmers after compensation they have become daily labourers, guards in cities, migrate to cities or they have continued to practice farming through land renting. This paper contributes to the existing literature on the effects of urban expansion and land expropriation on the livelihoods of peri-urban farmers by addressing several gaps in previous research. While there is an extensive body of literature on this topic (e.g., [1,3,4,10,15,19,21–26]) studies focusing specifically on North Wollo Zone remain scanty. For instance, [27] examined factors influencing the rezoning of Woldia city in the zone, while [28] explored the effects of land expropriation on livelihood changes in Kon and Gashena towns. However, these studies do not sufficiently capture the broader impact

of urban expansion and land expropriation on peri-urban farmers' livelihoods in the zone. Additionally, previous research, such as those by [26,29], has focused primarily on the determinants of livelihood diversification without considering the direct effects of expropriation on farmers' livelihoods. Similarly, [10] concentrated on the legality of land expropriation and compensation, while studies by [19,27,30] explored the economic linkages between urban development and peri-urban communities, but failed to examine the sectors evicted farmers engage in as part of their livelihood diversification strategies or how compensation affects short-term consumption patterns. Furthermore, much of the existing literature [1,22,26,29,31,32] has predominantly focused on the immediate impacts of land loss and compensation, without addressing the long-term sustainability of livelihood diversification strategies for displaced peri-urban farmers. Thus, this study fills these gaps by examining the effects of urban expansion and land expropriation on the livelihoods of peri-urban farmers in North Wollo Zone, Ethiopia.

## 2. Literature review

The world is undergoing rapid urbanization, particularly in developing countries like Ethiopia, where urban growth is accelerating [2,10]. While this trend is evident in Ethiopia, similar patterns and challenges are observed across Asia, Latin America, and sub-Saharan Africa. For instance, in China, large-scale land expropriation for urban expansion has led to widespread displacement of farmers, with studies showing that compensation often fails to match the long-term economic and social value of lost land, leading to reduced well-being and social unrest [14] and [13]. In Bangladesh, the political economy surrounding land acquisition reveals how peri-urban poor communities are disproportionately affected, with elite capture and limited participatory governance exacerbating vulnerability [33]. Latin American cases, such as in Colombia, further highlight how forced displacement due to urban megaprojects increases inequality and undermines the social fabric of peri-urban communities [34]. These international cases demonstrate the broader implications of urban land expropriation: inadequate compensation, disruption of traditional livelihoods, and social marginalization. Comparative literature suggests that effective policy responses must prioritize transparent compensation mechanisms, inclusive land governance, and comprehensive livelihood restoration programs. Without these safeguards, the pursuit of urban development risks entrenching poverty and exacerbating inequalities among peri-urban populations [35]. Urbanization's demand for land is often met by converting rural areas on the outskirts of cities, leading to the loss of agricultural land in peri-urban regions [10]. These areas, which are crucial to the livelihoods of farmers, are especially vulnerable to farmland loss due to urban expansion [32]. As a result, farmers in these areas face significant challenges, as their primary source of income is taken away. In developing countries, where most people live in highly concentrated peripheral areas and rely on fragmented landholdings, land expropriation poses a greater threat to livelihoods than in developed nations [10].

Urban expansion in Ethiopia, and other developing countries, has profound effects on farming communities, particularly in peri-urban areas. The conversion of agricultural land to urban use displaces farmers or even if they retain their residences, their income from farming is lost permanently, forcing them to seek alternative sources of income. However, as the economy shifts toward urban-focused industries and services, the traditional agricultural workforce often struggles to adapt [31]. The loss of productive land also reduces the supply of agricultural goods to urban areas, further impacting the affected households [1,8]. To overcome such vulnerabilities, the expropriated farmers plan livelihood diversification strategies [1]. Livelihood outcome is the result of strategies that are adopted by households in order to overcome the shock to build asset bases as opposed to poor livelihoods which deplete asset bases thereby increasing vulnerability [10]. The socio-economic well-being of people and households is directly impacted by the livelihood options they choose. Livelihood diversification is a common strategy used by displaced farmers, as they explore non-agricultural work, small businesses, or migration to urban centers [1]. While compensation payments, which is often part of land expropriation processes, are intended to offset the economic loss, they often fall short in providing long-term security, especially if beneficiaries lack the necessary skills or access to investment opportunities. It is also frequently insufficient to foster sustainable livelihoods in the long term, especially if beneficiaries lack the necessary skills, market access, or investment opportunities [19].

A growing body of empirical studies have documented the economic, social, and psychological impacts of land expropriation, such as income loss, community fragmentation, and mental health challenges [14–19]. Studies show that compensation, while intended to alleviate this economic burden, often falls short in replacing the long-term value of land and in supporting sustainable economic activities [10,31]. Furthermore, social and psychological effects are significant; the loss of land often leads to community disintegration, family disruption, and mental health issues. [14] finds that land expropriation in rural China leads to psychological distress and a decline in well-being, a finding echoed by [13], who highlight the health impacts of land loss on farmers in China. In Ethiopia, [1,10,19,31,32,36] report similar outcomes, noting that land loss can reduce social cohesion and exacerbate food insecurity, contributing to a sense of economic insecurity and psychological distress among peri-urban farming communities.

While compensation mechanisms like cash payments are common, their effectiveness has been a subject of debate. In Ethiopia, the Expropriation of Landholdings for Public Purposes and Payment of Compensation Proclamation [37] stipulates that landowners whose land is expropriated are entitled to compensation. However, studies suggest that compensation practices often do not fully meet the needs of affected farmers, especially when land is seen as more than just an economic asset, but also as a cultural and social asset [1,4,11]. Furthermore, the compensation process is frequently non-transparent, and the amount paid is often insufficient to restore lost livelihoods, particularly in peri-urban contexts where land values and living costs are rising [8,20]. Even with recent reforms aimed at improving compensation, the lack of adequate community participation and the absence of comprehensive livelihood restoration programs contribute to dissatisfaction among affected farmers [10,21].

Legal and policy frameworks surrounding land expropriation in Ethiopia have evolved to address some of these challenges. The Expropriation of Land for Public Purposes and Payment of Compensation Proclamation has undergone various revisions to enhance compensation and livelihood restoration for affected households. Despite legal frameworks for compensation, the implementation remains inconsistent, and many farmers find themselves unable to adapt successfully to new livelihoods. According to [8], the lack of a clear and robust policy framework for livelihood restoration and the absence of adequate rehabilitation programs have led to prolonged economic vulnerability for evicted farmers. These findings align with global experiences, underscoring the need for comprehensive policies that not only provide fair compensation but also support livelihood diversification and community resilience in the face of urban expansion. However, in many cases in developing countries including Ethiopia the plans to address development-related problems are not comprehensive and the effort to understand the livelihood strategies of expropriated farming households is very limited. This study examines the effect of urban expansion and land expropriation on the livelihoods of peri urban farmers in North Wollo Zone, Ethiopia.

### 2.1. Conceptual framework of the study

Fig 1 presents the conceptual framework of the study, which is adapted from [7], establishes the nexus between urban growth, peri-urban transformation, and the sustainable livelihoods of peri-urban residents, and is guided by the Sustainable Livelihoods Framework (SLF) [38]. It is particularly important for this study as it seeks to understand how urbanization, particularly through agricultural land use change and expropriation, creates a specific vulnerability context for peri-urban communities. Urban expansion results in increased population density and built-up areas, which in turn connect peri-urban economies more closely to urban centers. If managed effectively, this process can create job opportunities, improve access to services, and support the integration of peri-urban households into the broader urban economy. However, the outcomes of this transformation are not uniform.

Not all peri-urban residents possess the same set of livelihood assets, such as natural capital (land), human capital (education and skills), financial capital, physical infrastructure, and social capital (community networks), which significantly influences their capacity to adapt. Consequently, coping strategies vary, with some households able to diversify into non-farm employment, migrate, or establish small-scale enterprises, while others face heightened livelihood insecurity

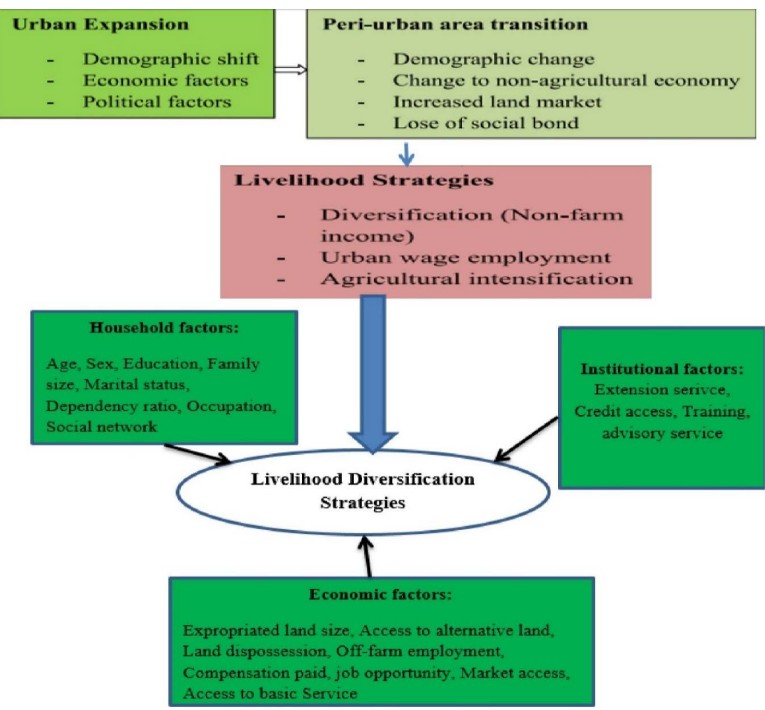

**Fig 1. Conceptual framework of the study.**

due to limited alternatives and inadequate compensation. This framework illustrates how urban expansion and the resulting peri-urban transition, driven by demographic, economic, and political factors, transform traditional rural livelihoods, prompting households to adopt diverse livelihood strategies such as non-farm income generation, urban wage employment, and agricultural intensification. These strategies are influenced by household-level factors (e.g., education, family size, occupation), economic factors (e.g., land expropriation, market access, employment opportunities), and institutional factors (e.g., extension services, credit access, training). As peri-urban areas shift from agrarian to more urban-oriented economies, the framework highlights the importance of adaptive livelihood diversification in enhancing household resilience, ensuring income security, and mitigating the socio-economic impacts of urbanization. Therefore, the study applies this framework to examine how urbanization and land expropriation affect the livelihood assets, strategies, and outcomes of peri-urban farmers in the North Wollo Zone of Ethiopia.

## 3. Research methodology

### 3.1. Study area profile

North Wollo Zone, located in the Amhara Region of Ethiopia, covers 12,172.50 square kilometers and has a population of over 1.8 million people [9]. Geographically, it spans between latitudes 11°49'59.99" N and longitudes 39°14'60.00" E, with elevations ranging from 913 to 4,187 meters above sea level, contributing to its diverse climate and topography. Woldia, the administrative center, sits at 2,112 meters above sea level. The zone is bordered by South Wollo to the south, Wag Hemra to the north, Tigray Region to the northeast, and Afar Region to the east. Agriculture, particularly the cultivation of teff, barley, wheat, and legumes, is the economic backbone of the zone. However, only 24% of the land is arable, and the region faces significant challenges such as soil degradation, food insecurity, and limited infrastructure [9]. Recent urbanization has led to increased land expropriation for investment, residential housing and infrastructure construction,

particularly around Woldia and other towns in the zone. The amount of land demanded for different urban development purposes in the zone is increasing every year and built-up areas increased as a result of horizontal expansion. This urban expansion impacts peri-urban farmers, both by providing new opportunities for market access and by displacing agricultural land [20]. Peri-urban farming in North Wollo, particularly around towns like Woldia, exhibits distinct features compared to other regions in Ethiopia [9]. The area's unique topography, climate, and proximity to urban centers present particular challenges and opportunities for local farmers [36]. These factors contribute to a unique socio-economic context that warrants specific study of the impacts of urbanization and land expropriation on livelihoods in the zone.

## 3.2. Research design

In line with the approach outlined by [39], the study employed a primarily quantitative research design, supplemented by qualitative insights, to investigate the effects of urbanization and land expropriation on the livelihoods of peri-urban farmers in North Wollo Zone. The rationale for using this approach is that it allows for a more comprehensive understanding of the research problem by capturing both quantitative and qualitative data from respondents and informants at a glance. The study first collected quantitative data through structured questionnaires distributed to a representative sample and the quantitative part generates numerical data that can be statistically analyzed, providing a broad, generalizable understanding of the issue based on a sample of respondents. To enhance and contextualize the quantitative findings, a limited qualitative component was included in the form of key informant interviews (KIIs) and focus group discussions (FGDs). These were used to capture select perceptions, lived experiences, and community narratives that helped illuminate specific patterns identified in the quantitative data. This approach mitigates the limitations of a single method by combining the strengths of both, enabling a deeper understanding of quantitative findings through participants' perspectives and experiences captured qualitatively [10].

## 3.3. Data description

To achieve the study objectives, the study collected both quantitative and qualitative data from primary sources. Data were collected from peri-urban farmers whose land had been expropriated and who received compensation over the past 10–20 years, as well as from key stakeholders involved in the processes of land expropriation, compensation, and rehabilitation. For the primary data collection, structured questionnaires were used to gather information from farmers about their demographics, socio-economic characteristics, asset holdings, and livelihood diversification strategies. The main aim of these questions was to capture farmers' opinions, viewpoints, and aspirations regarding how urban expansion affects the livelihoods of nearby agrarian communities. The questionnaire was pretested and revised as necessary before data collection began. To eliminate language barriers, the questionnaire was translated into the local language, Amharic, and then back into English for analysis. Additionally, face-to-face interviews were conducted with two farmer association leaders, two local land administration experts, one property valuation expert, and three expropriated farmers to explore the factors influencing livelihood diversification and the policies related to land expropriation and compensation. Four separate focus group discussions (FGDs), each involving 6–8 participants, were held to cross-check and validate the data collected through other methods. These discussions involved expropriated farmers, community leaders, land officials, and experts, and focused on the historical development of towns and their effects on local farming communities, offering deeper insights into the challenges faced by peri-urban farmers. The FGDs were conducted in collaboration with the city and towns of the Zone municipality officers. Throughout the data collection process, the researchers and selected officers closely supervised the fieldwork. Data collection took place from 16/03/2024–27/07/2024.

## 3.4. Sampling design

The target group for this study comprised peri-urban farmers in North Wollo, whose land was lost due to horizontal urban expansion, investment projects, and infrastructural development. These farmers, particularly heads of households, were

selected as respondents to represent the broader peri-urban farming community. The zone was chosen because it is one of the earliest centers of urbanization in Ethiopia, where rapid urban growth has significantly affected agricultural land use and farmers' livelihoods. A multistage sampling design was employed to select participants for the study. Initially, four cities of the zone, Woldia, Kobo, Mersa, and Lalibela were purposively selected based on their high rates of urbanization and industrial investment, which have notably impacted land use and farmers' income sources. In the second stage, peri-urban farmers whose land had been expropriated and compensated due to urban expansion were specifically targeted. According to these cities municipal administration report, a total of 3,520 expropriated farmers were compensated for their land loss as a result of urbanization and industrial development from 2010–2024 [9]. Snowball sampling was then used to select households that had been displaced by land expropriation. Finally, the sample size for each city was determined using proportional sampling, ensuring that the sample represented the population of each city accurately. Using the [40] formula with a 5% precision level, the overall sample size was calculated as 378 farm households, and a simple random sampling method was used to select the participants.

$$n = \frac{N}{1 + N(e^2)} \qquad n = \frac{3520}{1 + 3520(0.0025)} = 378$$

Whereas: n is the sample size; N is the population size, and e is the level of precision.

The interviews with key informants and the focus group discussions provided in-depth insights by engaging with individuals who were knowledgeable and relevant to the topic. These methods also helped the researcher's cross-check and validate the data collected from the questionnaires, ensuring the reliability and consistency of the findings. Recognizing potential biases in key informant interviews and focus group discussions, particularly from purposive sampling, is crucial. To mitigate this, the study triangulates data from questionnaires, interviews, and focus groups to enhance reliability and validity through cross-checking consistency [10].

## 3.5. Methods of data analysis

Data for this study were coded and processed using STATA statistical software. The analysis included both descriptive and inferential methods. Descriptive statistics such as means, standard deviations, tables, and figures were used to summarize the data. For the inferential analysis, a binary logistic regression model was employed to estimate the factors influencing farmers' livelihood diversification decisions. Qualitative data from key informant interviews and focus group discussions were analyzed thematically and synthesized.

## 3.6. Econometric model specification

In this study, the dependent variable, livelihood diversification status, is dichotomous: it takes a value of 1 if a household diversifies its livelihood and 0 if it does not. Given this binary outcome, two suitable models are available for analysis: the logit and probit models. While both models can be used to estimate probabilities, the logit model is generally simpler to estimate and tends to produce more stable results compared to the probit model [41]. As a result, we used the logit model to estimate the results of the study. Specifically, our aim is to estimate the probability that a household diversifies its livelihood, given the set of explanatory variables.

Previous studies (e.g., [31,32,36,42,43] and [44]) have revealed that the livelihood diversification of peri-urban farm households, driven by urban expansion and land expropriation, is significantly influenced by a range of factors. These include capital endowments (human, physical, and financial), socio-economic conditions, adult labor, local market size, past experience in the non-farm sector, location, and external shocks. Thus, a household's decision to diversify depends on its demographic characteristics, asset holdings, past diversification experience, exposure to shocks, and the local economy.

Following [41,45] random utility framework, the model assumes that a rational household aims to maximize utility by comparing the benefits of different livelihood strategies. The household will choose the strategy that yields the highest net benefit, thereby maximizing its utility from a set of discrete alternatives.

$$U_{ij}^* = F_i(X_{ij}\beta) + \mu_{ij} \tag{1}$$

Where; $U_{ij}^*$ is the latent variable representing the net benefits a diversification strategy, $X_{ij}$ includes the household specific characteristics, capitals (human, physical and financial) endowments, size of the local market, past experience in the non-farm sector, location and shocks, $\beta$ is parameter to be estimated, and $\mu_{ij}$ is the error term. The probability of a household diversifying its livelihood is:

$$D_i^* = X_i + \S \; ; \; \text{with} D_i = \begin{cases} 1 & \text{if } D_i^* > 0 \\ 0 & \text{otherwise} \end{cases} \tag{2}$$

$$P_i = \Pr(D_i = 1|X_i) = \frac{e^{X_i\beta}}{1 + e^{X_i\beta}} \tag{3}$$

Where $D_i = 1$ indicates diversification, and $X_i$ represents the set of explanatory variables. Following [41], the logit model is expressed as:

$$L_i = \ln\left(\frac{P(Y_i = 1/X_i)}{1 - P(Y_i = 1/X_i)}\right) = Z_i = \alpha + X_i\beta + \varepsilon_i \tag{4}$$

Where $Z_i$ represents the dependent variable (household livelihood diversification status). $\alpha$ is the intercept of the model, and $\ss_i$ represents the unknown parameters to be estimated. $X_i$ is a set of explanatory variables (farm household specific characteristics, institutional factors, socio-economic factors, and external factors) that influence the dependent variable, and $\varepsilon_i$ is the error term of the model. It is important to note that the estimated coefficients do not directly show the effect of changes in the corresponding explanatory variables. Therefore, the study estimates marginal effects to capture the effect of changes in the explanatory variables on the probability (P) of the outcome occurring.

The marginal effect of each variable on the probability of diversification is calculated using:

$$\frac{\partial P(D = 1|X)}{\partial X_j} = P(D = 1 | X) \times (1 - P(D = 1 | X)) \times \beta_j \tag{5}$$

Maximum likelihood estimation is used to estimate the model. This approach helps identify the key factors influencing farmers' decisions to diversify their livelihoods after land expropriation and compensation.

According to [41] and [1], logistic regression assumes no multicollinearity among the independent variables, meaning they should not be highly correlated. In this study, multicollinearity was assessed using the variance inflation factor (VIF), with all variables showing a VIF below 10, indicating no issues of multicollinearity. The dependent variable, livelihood diversification status, is binary (diversified/not diversified), fulfilling the binary requirement for logistic regression. Additionally, logistic regression requires a sufficient sample size, with a minimum of 10 cases per independent variable. Given 22 variables in this study, the minimum required sample size was 220, and the study used 378, exceeding this requirement. Lastly, logistic regression assumes no heteroscedasticity or non-normality, and robust regression techniques were applied to address these potential issues in the analysis. Table 1 provides the description, measurement and hypothesis of the study.

**Table 1. Description, measurement and hypothesis for the selection variables.**

| Dependent variable for the outcome equation: Livelihood diversification (= 1 for yes and 0 otherwise) | | |
|---|---|---|
| **Explanatory Variables** | **Description and measurement** | **Expected sign** |
| Gender | Gender of the household head (Female as reference) | +/- |
| Marital status | Marital status of the household head (Single as reference) | +/- |
| Education | Education level of the household head, measured in schooling years | + |
| Age | Age of the household head, measured in years | +/- |
| Dependency ratio | The ratio of unemployed to labor force, measured in ratio | – |
| Family size | Total number of household members | +/- |
| Expropriated Land size | Size of expropriated land (in hectare) | – |
| Off-farm Employment | =1 If the household is engaged in off farm activities, 0 otherwise | + |
| Compensation paid | =1 If the household received fair compensation for their expropriated land, 0 otherwise | + |
| Extension service | =1 If the household has access to extension service after expropriation and compensation, 0 otherwise | + |
| Market access | =1 If the household has access to market access after expropriation and compensation, 0 otherwise | + |
| Occupation | =1 If the household's main job is farming, 0 otherwise | – |
| Land dispossession | =1 If the household's land is expropriated fully, 0 otherwise | – |
| Access to alternative land | =1 If the household has access to alternative land resource, 0 otherwise | + |
| Training access | =1 If the household gets training access after land expropraion and compensation, 0 otherwise | + |
| Credit access | =1 If the household has access to credit, 0 otherwise | + |
| Social network | =1 If the household was able to mantainn or expand their social networks after land expropriation, 0 otherwise | + |
| TLU | Livestock asset measured by TLU | + |

## 3.7. Ethical consideration

Ethical approval for this study was obtained on 15/03/2024 from the Research Ethics Approval Committee of Woldia University, Ethiopia, and authorized by the Institutional Review Board (IRB) of the College of Business and Economics at Woldia University under Ref. No: FBE/RCSTT/247/2024. All participants provided written informed consent, with explicit acknowledgment that participation was voluntary and they had the right to withdraw at any time without consequence. Participants were also informed that their data would be used solely for analyzing the effects of urbanization and land expropriation on the livelihoods of peri-urban farmers in North Wollo Zone, Ethiopia, and that their responses would be kept confidential and shared anonymously. Before completing the questionnaire or participating in interviews or focus group discussions, the purpose of the research was clearly explained to ensure understanding. Additionally, all data analyzed in the study was de-identified to prevent any risk of participant identification.

## 4. Results and discussion

### 4.1. Descriptive analysis

The summary statistics presented in Table 2 provide valuable insights into the livelihood diversification strategies employed by peri-urban farmers in the wake of land expropriation and compensation. Out of the 378 respondents, 96.6% (372 households) provided complete responses. The majority of household heads were male (69.09%), and most households (86.02%) were headed by married individuals. Farming remained the primary occupation for 80.38% of households, with 67.74% engaged in off-farm employment, indicating a growing trend toward livelihood diversification. Farming is the primary occupation for most households (80.38%), while the remaining 19.62% are engaged in other types of work. While 44.09% of households have maintained or expanded their social networks, the majority (55.91%) have not. Discussants noted that households who were fully expropriated and relocated are still in the process of integrating into their new

**Table 2. Descriptive summary of categorical variables.**

| Variables | Category | Frequency | Percent |
|---|---|---|---|
| Gender of the household head | Male | 257 | 69.09 |
| | Female | 115 | 30.91 |
| Marital status | Single | 10 | 2.69 |
| | Married | 320 | 86.02 |
| | Separated | 26 | 6.99 |
| | Widowed | 16 | 4.30 |
| Occupation | Farming | 299 | 80.38 |
| | Other | 73 | 19.62 |
| Land dispossession | Fully | 123 | 33.06 |
| | Partly | 249 | 66.94 |
| Access to alternative land | Yes | 282 | 75.81 |
| | No | 90 | 24.19 |
| Livelihood diversification | Yes | 228 | 61.29 |
| | No | 144 | 38.71 |
| Compensation Receiver from the family | Male head | 224 | 60.22 |
| | Female head | 102 | 27.41 |
| | Both male and female head | 46 | 12.37 |
| Off-farm employment | Yes | 252 | 67.74 |
| | No | 120 | 32.26 |
| Training access | Yes | 71 | 19.09 |
| | No | 301 | 80.91 |
| Market access | Yes | 240 | 64.52 |
| | No | 132 | 35.48 |
| Compensation paid | Fair | 36 | 9.68 |
| | Not fair | 336 | 90.32 |
| Extension service | Yes | 137 | 36.83 |
| | No | 235 | 63.17 |
| Credit access | Yes | 140 | 37.63 |
| | No | 232 | 62.37 |
| Social network maintained/expanded | Yes | 164 | 44.09 |
| | No | 208 | 55.91 |

Source: Own survey, 2024.

neighborhoods, with many yet to join local associations. Additionally, only 36.83% had access to extension services, and 64.52% had market access, reflecting limited support for expanding income sources. Compensation was predominantly received by male heads (60.22%), and 61.29% of households diversified their livelihoods, largely in response to land expropriation. A significant portion (66.94%) faced partial land loss, and 75.81% had access to alternative land, which facilitated adaptation. Yet, barriers such as limited access to extension services (36.83%) and credit (37.63%) hindered the full potential of diversification efforts. KII and FGD discussants noted that households were not given the opportunity to seek legal advice or participate in public discussions before the expropriation, as the municipality directly took their farmland.

Furthermore, 90.32% of households felt the compensation was unfair, citing insufficient value and a lack of guidance on how to utilize the funds. The expropriation process was criticized for not involving public consultation and many believed it

was driven by administrative interests rather than public benefit. Most discussants felt the compensation was neither fair nor reflective of the actual value of the expropriated land. They highlighted that the government often takes land with the justification of "land is for the government" but compensates only with cash, without offering guidance on how to manage or invest the money. Furthermore, there was no follow-up from authorities after compensation. The discussants also argued that the expropriation was not driven by a genuine public purpose but rather by the interests of the administrative body, suggesting that such actions should not primarily serve the commercial interests of the state or private entities. Moreover, according to the interviewees, the main reason for their dissatisfaction was the none-participatory practices in the determination of compensation amounts.

Finally, the summary shows that 61.29% of households in the sample have diversified their livelihoods, indicating that most affected farmers have adapted to land dispossession by seeking alternative sources of income beyond traditional farming. The high rate of off-farm employment, with 67.74% of households having members involved in such activities, further highlights this shift towards diversification. While many households have attempted to diversify their income sources, the data also highlights ongoing challenges and constraints that could undermine the long-term sustainability of these adaptations.

Table 3 summarizes key continuous variables that provide insights into the livelihood diversification strategies of peri-urban farmers affected by land expropriation. The average age of household heads (49.16 years) suggests a relatively experienced population, potentially aiding their adaptation to land dispossession. With an average of 5.28 years of education, these farmers may have a moderate capacity to diversify into non-agricultural income sources. The average family size of 4.90 members indicates available labor for diversified activities, although a high dependency ratio of 0.838 suggests that many household members are economically dependent, limiting labor flexibility. The average expropriated land size of 0.618 hectares highlights the significant loss of productive assets, likely driving the need for alternative income sources, as evidenced by the 67.74% of households involved in off-farm employment and 61.29% diversifying their livelihoods. The average TLU of 7.66 suggests livestock plays a role in diversification, offering resilience against land loss.

## 4.2. Causes of urban expansion

Fig 2 presents the relative weight of causes of urban expansion in the study area. Understanding these causes is crucial for managing urban growth and its impacts. The survey revealed that all respondents agreed the town had rapidly expanded into surrounding rural kebeles, with urban growth primarily encroaching on agricultural land. The key drivers identified were migration (41.9%), natural population increase (23.6%), reclassification of rural kebeles to urban status (18.9%), and urban investment projects (15.6%). The expansion of the city has led to the incorporation of adjacent rural areas to meet the demand for land for industrial corridors, housing, and infrastructure. Additionally, all key informant interviewees and municipal officials reported towns like Woldia, Kobo, Mersa, and Lalibela have experienced rapid expansion

**Table 3. Descriptive summary of continuous variables.**

| Variables | Mean | Std. deviation | Minimum | Maximum |
|---|---|---|---|---|
| Age of the household head | 49.16 | 9.71 | 26 | 78 |
| Education of the household head | 5.28 | 3.59 | 0 | 17 |
| Family size | 4.90 | 1.48 | 2 | 8 |
| Dependency ratio in rate | .838 | .217 | 0 | 1.67 |
| Expropriated land size in hectare | .618 | .415 | .25 | 1.5 |
| TLU | 7.66 | 4.59 | 0 | 26.49 |

Source: own Survey, 2024.

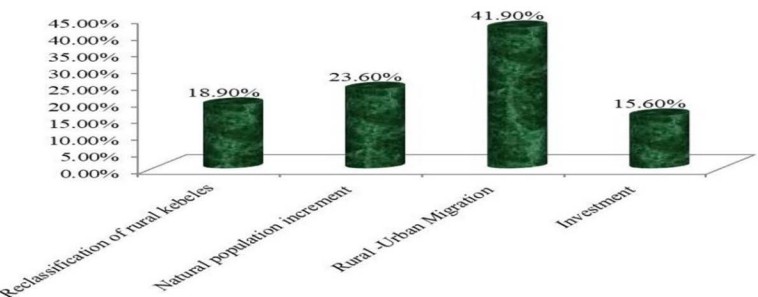

**Fig 2. Relative weight of causes of urban expansion.**

due to their designation as key urban centers and their involvement in zonal development plans. This growth has led to the incorporation of surrounding rural areas to meet land demands for industrial, housing, and infrastructure projects, significantly contributing to population increases beyond national trends. Interviews and discussions indicated that this rapid urban growth has resulted in the expropriation of farmland from peri-urban farmers, significantly impacting their livelihoods. Local authorities have been acquiring land for residential, commercial, and infrastructure development, leaving little time for affected farmers to adapt or diversify their income sources. This has created a major challenge for peri-urban communities, who are facing the sudden loss of their primary agricultural resources. These findings resonate with broader concerns in the literature on peri-urban transformation, where governance failures, such as lack of consultation, inadequate compensation, and absence of livelihood planning, undermine household resilience and exacerbate socio-economic vulnerability. The case highlights a critical tension between urban development imperatives and equitable land governance, raising questions about how state-led urbanization can better integrate inclusive and sustainable planning frameworks.

### 4.3. Use of compensation payment

Table 4 provides an overview of how households utilized compensation payments received due to land expropriation. The majority (84.13%) of households spent the funds on housing and household items, such as mobile phones, TVs, refrigerators, and furniture, indicating a focus on improving living standards rather than economic investment. A significant proportion also invested in income-generating assets, with 40.59% purchasing Bajajis (three-wheeled vehicles) and 12.63% acquiring animal transport carts. Around 20.96% started small businesses, and 8.87% invested in agricultural machinery. Additionally, 36.82% deposited the compensation in banks, while 47.31% invested in their children's future. The data also reveals that 21.77% of households used the compensation to build or repair homes, 18.54% paid off debts, and 11.02% allocated the funds for other purposes. Discussants noted that many households, particularly those with lower education levels, lacked knowledge on how to best utilize the compensation, often opting for safe investments like bank deposits. Furthermore, households faced food insecurity, social disruptions, and family disintegration following relocation, with some using the compensation to cover daily consumption needs. Overall, the compensation was used for a range of purposes, from satisfying immediate needs to strategic investments, reflecting the diverse priorities and challenges faced by the affected households in the context of land expropriation.

### 4.4. Effect of land compensation on consumption patterns

Table 5 highlights that the majority (80.91%) of household's experienced short-term consumption changes following compensation payments, indicating financial pressures that may limit long-term livelihood diversification. Specifically,

**Table 4. Use of compensation payment.**

| Indicators | Frequency | Percent |
|---|---|---|
| Building and repairing house | 81 | 21.77 |
| Housing facility (mobile, TV, refrigerator, Sofa, washing machine, stove, air-condition, motor bike, laptop, furniture etc) | 313 | 84.13 |
| Buying Bajaji | 151 | 40.59 |
| Gari –cart | 47 | 12.63 |
| Investing in agri-machines (sprayer, hoes, cattle, tresher, tractor) | 33 | 8.87 |
| Small business | 78 | 20.96 |
| Fiat for rent | 17 | 4.56 |
| Deposit currency in the bank | 137 | 36.82 |
| Paying debts | 69 | 18.54 |
| Investment for children | 176 | 47.31 |
| Others (gift for other family) | 41 | 11.02 |

Source: own survey, 2024.

**Table 5. Consumption change after land expropriation.**

| There is short run consumption change after compensation | Yes | 301 | 80.91 |
|---|---|---|---|
| | No | 101 | 19.09 |

Source: own Survey, 2024.

compensation was spent on food, particularly meat and alcohol, as well as clothing, recreational activities, and ceremonies. These shifts suggest that the compensation provided immediate financial relief and increased purchasing power. The discussants emphasized that compensation led to a short-term shift in consumption patterns among farm households. The compensation money was primarily spent on higher-value food items, particularly meat and alcohol, signaling a change in dietary preferences. In addition, households allocated funds for new clothing, suggesting a desire for personal or lifestyle improvements. There was also a noticeable increase in spending on recreational activities, highlighting a focus on leisure and social enjoyment. Moreover, a portion of the compensation was directed towards ceremonial purposes. Overall, the compensation resulted in a shift towards indulgent and social spending, illustrating how such financial windfalls can rapidly influence household priorities.

However, this short-term consumption boost may not translate into long-term resilience or sustainable income diversification. The allocation of compensation towards immediate consumption, rather than productive investments, could hinder households' ability to establish alternative income sources or recover from land loss in the long run. Therefore, it is crucial to understand how these consumption changes affect household finances and livelihood decisions. Targeted support and monitoring may be needed to help redirect compensation towards productive investments that promote long-term resilience in families of expropriated household's.

## 4.5. Post-expropriation challenges for farm households

Fig 3 presents the challenges that farm households faced after expropriation and compensation, with particular emphasis on factors that hindered their ability to diversify livelihoods. A large proportion (89.24%) of respondents identified inflation as a major challenge, which eroded their purchasing power, devalued assets, and made it more difficult to invest in new income sources, increasing overall vulnerability. The impact of inflation is especially concerning, as it erodes the ability

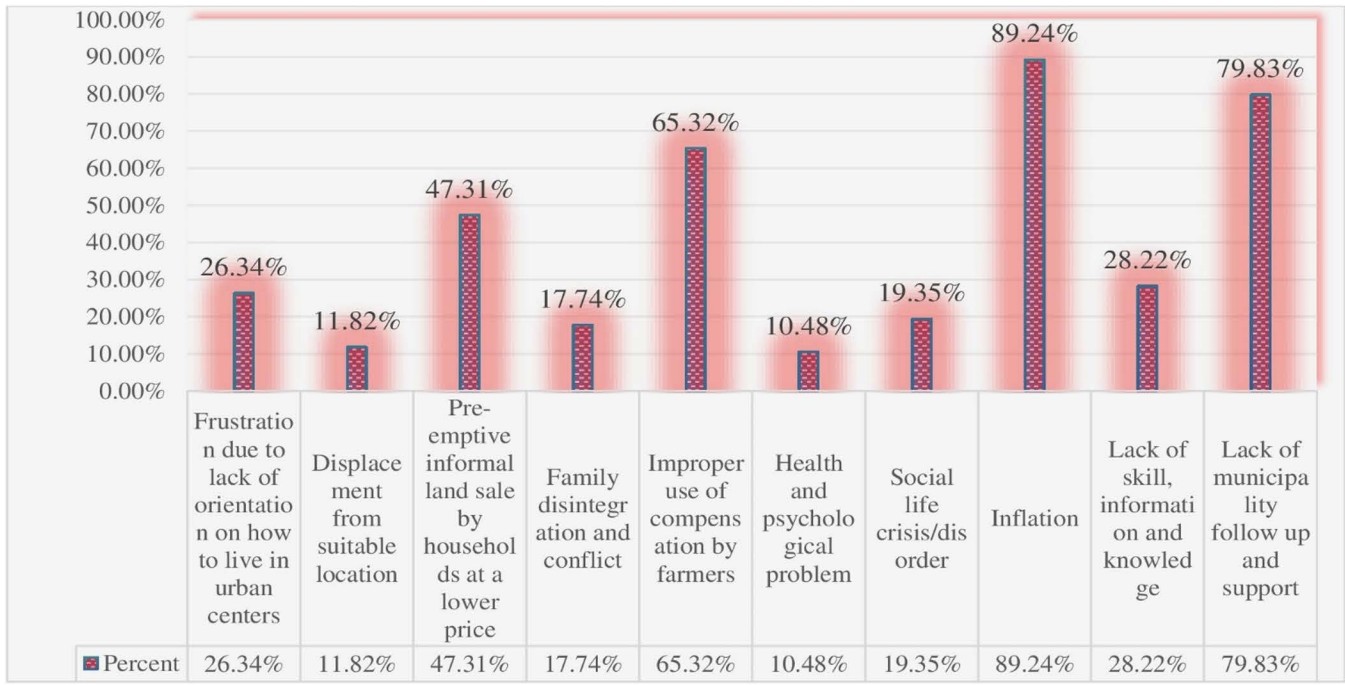

**Fig 3. Challenges that farm households faced after expropriation and compensation.**

of households to build long-term resilience, compounding their vulnerability after losing their land. Another notable challenge, identified by 26.34% of the respondents, was the lack of guidance on adapting to urban life. Transitioning from rural to urban areas is inherently difficult, and many households were ill-equipped to navigate urban environments. The absence of knowledge about urban living skills, such as accessing services, employment opportunities, and adjusting to urban economies, limits their capacity to diversify income sources. The high inflation and lack of urban living skills critically undermine efforts for livelihood diversification. These findings suggest that merely providing compensation is insufficient to support long-term resilience for expropriated households [10,26].

Moreover, 47.31% of respondents reported engaging in informal land sales at lower prices due to pressure, which often leads to poor long-term outcomes. While these informal transactions might seem advantageous in the short term, they undermine the ability to make strategic investments and contribute to the growth of informal settlements, negatively impacting both individual livelihoods and broader urban development. As noted by [1], when land is reclassified as urban, farmers often resort to informal sales to avoid losses from inadequate government compensation, even though this results in long-term harm to their economic stability. Additionally, informal land sales contribute to the growth of informal settlements, negatively impacting urban development. Ultimately, government expropriation without fair compensation forces farmers into vulnerable land markets, undermining their well-being. Furthermore, all FGD discussants, key informant interviewees and municipal officials informed the prevalence of massive informal land sales and construction in all towns of the zone.

Additionally, 65.32% of households struggled with improper use of compensation funds, which hindered their capacity to invest in sustainable livelihoods. The issue stems from several factors, including low financial literacy and a lack of guidance on managing compensation. Many households, receiving lump-sum payments, lacked the knowledge or resources to allocate these funds for long-term stability. Without proper financial education or support, families may have prioritized short-term needs over sustainable investments. Additionally, the pressure to adapt to urban living and cover daily expenses may have led to improper fund use, hindering long-term livelihood diversification [10,46].

Other challenges, such as displacement from suitable locations (11.82%), family disintegration and conflict (17.74%), social disruption (19.35%), and lack of skills and knowledge (28.22%), further complicate the transition process. These challenges indicate the multifaceted nature of the issue, with economic, social, and psychological factors all playing significant roles. Displacement often forces households to relocate to areas with limited access to essential services, such as healthcare, education, and transportation, making it difficult to establish stable livelihoods. Family disintegration may result from the stress and upheaval caused by land expropriation, leading to weakened family structures and reduced social cohesion, which can, in turn, affect collective coping strategies and resource sharing. Social disruption, including the breakdown of established community ties and social networks, impedes the ability of displaced households to adapt to urban settings, as these networks often provide crucial support during transitions [1]. Additionally, the lack of skills makes it harder for expropriated farmers to engage in alternative livelihood activities, especially in urban environments where specialized knowledge or vocational training is often required [15].

Taken together, these findings highlight the need to approach land expropriation not only as a legal or economic process but also as a governance issue with deep social implications. Comparative experiences from other developing contexts show that successful transitions often depend on institutional support mechanisms, such as vocational training, financial literacy programs, and post-compensation follow-up, which are currently lacking in the study area. Embedding expropriation and compensation processes within a broader framework of livelihood resilience and participatory urban governance is therefore essential to mitigating long-term vulnerability [14].

## 4.6. Livelihood copying strategies

The continued expansion of urban areas into rural regions and the growing interest of municipal governments in land conversion have led to inevitable changes in livelihoods for those who once relied on natural resources. As a result, people living on the urban periphery must adopt various survival strategies to cope with these changes. Households may turn to farm-based solutions, non-farm activities, or a combination of both, depending on their circumstances [1,10]. Table 6 reveals that agriculture remains the dominant occupation, with many households continuing to engage in farming activities like crop production (31.18%), animal fattening (22.04%), poultry farming (22.58%), and dairy farming (11.82%). Some households have also resorted to renting out their land (10.21%) or working as agricultural laborers (4.30%).

In addition to agriculture, households have diversified into other sectors. In the service sector, many have sought formal employment (7.25%) or engaged in transportation (44.89%), café and restaurant operations (18.01%), and advertising and printing (9.94%). In trade, common activities include cereal trading (19.62%), liquor sales (33.87%), and operating petty shops (27.41%). These shifts were driven by the difficulty in acquiring land due to expropriation, forcing many to buy agricultural products from nearby rural areas and sell them in urban markets. However, despite these efforts, their living standards and income continue to decline.

In the manufacturing sector, households have turned to metal and woodwork (29.03%), as well as producing stoves, weaving, and spinning (6.18%). Some have also ventured into construction (12.36%) and mining (6.72%) to diversify their livelihoods. In addition to farming, some households have taken to working in sand and stone quarries as income from agriculture decreases. While most households initially relied on farming, such as crop cultivation and animal husbandry, they have increasingly sought supplementary livelihoods in services, trade, manufacturing, and construction. However, the loss of land due to expropriation has forced them to rent land, but this solution has not been effective, as many farmers are unfamiliar with non-agricultural forms of subsistence. The diverse coping strategies adopted by these households highlight their resilience in the face of significant disruption to their traditional livelihoods.

Despite these efforts, there is a widespread lack of employment in the community, exacerbated by insufficient government compensation for expropriated land and the absence of a livelihood restoration program. According to the KIIs and FGDs, most farm households relied on agriculture, with supplementary activities in trade, services, and manufacturing. The expropriation process disrupted their traditional livelihoods, forcing them to explore a wide range of new economic

**Table 6. Livelihood diversification or coping strategies after expropriation.**

| Economic sectors | Sub group | Frequency | Percent |
|---|---|---|---|
| Agriculture | Crop production | 116 | 31.18 |
| | Sheep and goats | 68 | 18.27 |
| | Dairy farming | 44 | 11.82 |
| | Animal Fattening | 82 | 22.04 |
| | Honey production | 11 | 2.95 |
| | Vegetable and Fruit | 57 | 15.32 |
| | Poultry | 84 | 22.58 |
| | Rent farm land | 38 | 10.21 |
| | Agricultural employment and related activities (i.e., Labour payment in kind/ harvest share system; Daily labour in local area) | 16 | 4.30 |
| | Sell of home-wood, firewood, charcoal, grass | 33 | 8.87 |
| Service | Formal employee | 27 | 7.25 |
| | Illegal broker | 21 | 5.64 |
| | Cafe and restaurants | 67 | 18.01 |
| Tra | Transport | 167 | 44.89 |
| | Advertising and printing | 37 | 9.94 |
| | Car wash | 15 | 4.03 |
| Trade | Cereal trading | 73 | 19.62 |
| | Liquor sale (drinks) | 126 | 33.87 |
| | Domestic wholesale | 42 | 11.29 |
| | Petty shops/trade | 102 | 27.41 |
| | Vegetable and fruits sale | 37 | 9.94 |
| | Livestock sale | 89 | 23.92 |
| Manufacturing | Metal and wood work | 108 | 29.03 |
| | Stove, Electric *"mitad"*, weaving/spinning | 23 | 6.18 |
| | Grinding mills | 11 | 2.95 |
| | Repair service | 8 | 2.15 |
| Construction | Construction (concrete, gravel and general construction) | 46 | 12.36 |
| Mining | Mining (stone, sand, and other selected materials) | 25 | 6.72 |

Source: own Survey, 2024.

activities. After losing their land to expropriation, farmers were forced to rent land or engage in trade, manufacturing, construction, and service sectors. However, these alternatives were largely unsuccessful, as they lacked experience with anything other than agriculture for their livelihood. Discussants of the KIIs and FGDs expressed frustration over the lack of local employment and the absence of government compensation or livelihood restoration programs. As a result, many heads of households migrated to urban areas in search of work, leading to family disintegration as they were uprooted from their communities. This finding aligns with [1], who note that many displaced households lack the skills for urban jobs, forcing some heads of households to migrate to cities for work, often leading to family disintegration.

These findings reinforce broader concerns in the literature about the inadequacy of compensation-driven resettlement approaches, which often overlook the long-term livelihood needs of displaced communities. Comparative evidence from other rapidly urbanizing regions suggests that successful transitions require not only financial compensation but also comprehensive livelihood restoration planning, including skill development, employment facilitation, and institutional follow-up. The absence of such governance frameworks in the study area reveals critical gaps in urban policy implementation and raises important questions about the equity and sustainability of urban expansion processes [10,33,34,47].

Table 7 shows that 65.86% of respondents received residential plots during land expropriation, while 34.14% did not. In traditional farming communities, it is customary for all children to receive residential plots near their parents' home, along with separate farm plots. This system supports intergenerational land transfer and livelihood diversification. However, the lack of residential plots for 34.14% of respondents may hinder their ability to establish permanent homes, diversify income, and ensure the next generation's economic independence. This gap in land provision can significantly affect the well-being and livelihood strategies of these households.

## 4.7. Factors in choosing livelihood after expropriation

Fig 4 presents reasons for selecting a livelihood source after expropriation, which provides valuable insights into the decision-making processes of farm households in choosing their livelihood strategies following land expropriation. The most common reason, cited by 38.17% of respondents, was a focus on asset creation for tomorrow, indicating a forward-looking approach aimed at ensuring long-term economic security and resilience. This suggests that a significant portion of households were strategically planning their livelihood diversification to secure a more sustainable future. Additionally, 29.30% were influenced by observing the success of others in their community, highlighting the role of social learning and peer influence in shaping their decisions. A further 18.01% relied on their personal experience, emphasizing the importance of existing skills and expertise in guiding their choices. While a smaller proportion, 8.60% cited unspecified problems as a factor, the overall trend points to a deliberate and strategic approach to livelihood diversification among the majority of households affected by land expropriation. In an in-depth interview, one respondent shared the following insights: "I purchased a Bajaj after seeing my neighbors doing the same, without any government support or training. I attempted to work with it, but the effort was unsuccessful. Additionally, I cultivate crops on the land left after expropriation to help meet my family's basic needs. However, as urban expansion continues, farming job opportunities will gradually decline, reducing the prospects for those who rely on traditional farming skills to earn extra income." (In-depth interview with three household heads, all men over 39 years old, in Kobo Town, 2024).

**Table 7. Farm households who get urban land in addition to the compensation.**

| Q.Whether residential plots were given to each family member or not in addition to the compensation? | | Frequency | Percent |
|---|---|---|---|
| Response | Yes | 245 | 65.86 |
| | No | 127 | 34.14 |

Source: own Survey, 2024.

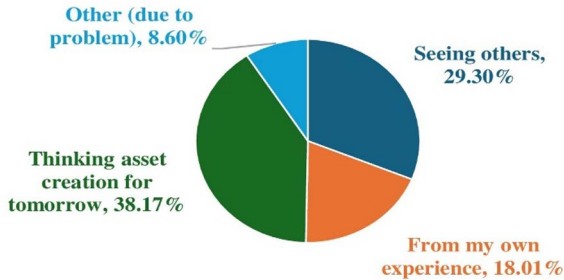

**Fig 4. Reasons for selecting a livelihood source after expropriation.**

## 4.8. Future livelihood outlooks

Fig 5 presents on the perceived future livelihood outlooks of farm households, due to land expropriation and low compensation, reveals a concerning picture of apprehension and uncertainty about the future. The most prominent sentiment, expressed by 52.95% of respondents, is that "it will be difficult" for them to maintain their livelihoods and economic well-being moving forward. This reflects a widespread sense of fear and uncertainty regarding their ability to overcome the challenges posed by land loss and inadequate compensation. Additionally, 38.70% perceive their future as "worst," indicating a deep pessimism about their economic security and living standards deteriorating as a result of the expropriation.

While a small minority, 6.18%, expressed a "good" outlook and 3.76% considered their future the "best," these positive sentiments were in the clear minority. Furthermore, 32.52% of respondents felt that their situation would "somewhat get lower," showing cautious optimism but acknowledging the challenges ahead. Notably, 15.86% were uncertain and stated they "can't say anything" about their future, underscoring the prevailing sense of anxiety and unpredictability among these households. Data from the discussants confirm the quantitative findings, highlighting the limited livelihood strategies in the study area, which are contributing to widespread unemployment, food insecurity, and poverty. Agriculture remains the primary livelihood strategy, with land cultivation being the main source of income, while animal husbandry serves as a supplementary activity. However, after losing their land due to expropriation, farmers were forced to rent land from others, which proved largely unsuccessful. This shift not only deepens unemployment but also exacerbates food insecurity, as many are unable to produce enough food to meet their needs. The discussants pointed out that they are not familiar with alternative means of subsistence, trapping them in a cycle of poverty. Regarding their future outlooks, the discussants expressed similar sentiments of concern and uncertainty, with most fearing that their situation would worsen in the coming years. A few, however, held onto cautious hope, believing that with the right support, they might find ways to improve their circumstances, though they emphasized the lack of clear avenues for doing so.

## 4.9. Empirical analysis

The results from the binary logit model are presented in Table 8. Prior to analysis, the data were checked for outliers, and a preliminary descriptive analysis was conducted to identify and correct any inconsistencies. Several diagnostic tests were performed to ensure data reliability and the model's suitability. The model fits the data well. The Wald test indicated that the null hypothesis (all regression coefficients equal to zero) was rejected with [$\chi^2(22) = (147.59)$; $P > 0.000$]. Additionally, the Variance Inflation Factor (VIF) for continuous variables and the correlation coefficient for categorical variables confirmed no significant multicollinearity issues. Finally, robust regression was applied to address potential heteroscedasticity and non-normality problems. The estimated results highlight the factors influencing the livelihood diversification strategies of peri-urban farm households after land expropriation and compensation in the North Wollo Zone, Ethiopia.

The results of the study indicate that the education level of the household head has a positive and statistically significant impact on the likelihood of households adopting livelihood diversification strategies after land expropriation and compensation, at the 5% significance level. This suggests that higher levels of education enhance the household head's

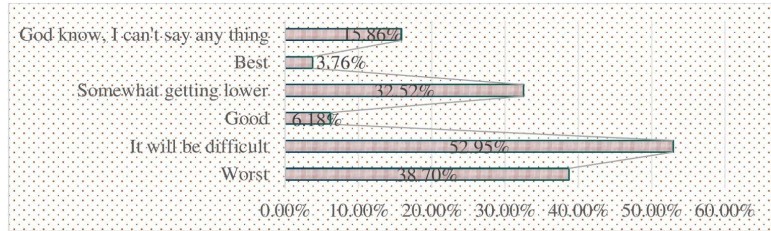

**Fig 5. Perceived future livelihood outlooks.**

**Table 8. Estimated results of binary logistic regression.**

| Livelihood diversification | Coef. | St.Err. | t-value | p-value | Sig | dy/dx |
|---|---|---|---|---|---|---|
| Gender of the household | −0.055 | 0.268 | −0.21 | 0.837 | | −0.012* |
| Age of the household | 0.005 | 0.014 | 0.39 | 0.699 | | 0.001 |
| Education of the household | 1.460 | 0.649 | 2.25 | 0.024 | ** | 0.134 |
| Marital status: base (single) | | | | | | |
| Married | 0.939 | 0.475 | 1.98 | 0.048 | ** | 0.104* |
| Divorced | 1.129 | 0.880 | 1.28 | 0.200 | | 0.220* |
| Widowed | 0.676 | 0.966 | 0.70 | 0.484 | | 0.142* |
| Occupation | −0.064 | 0.021 | −2.93 | 0.003 | *** | 0.035* |
| Household size | −1.398 | 0.772 | −1.81 | 0.070 | * | −0.006 |
| Dependency ratio | 0.489 | 0.561 | 0.87 | 0.383 | | 0.114 |
| Expropriated land size | −1.310 | 0.324 | −4.04 | 0.000 | *** | −0.108 |
| Land dispossession | −0.371 | 0.231 | −1.61 | 0.108 | | −0.047* |
| Access to alternative land | 1.326 | 0.513 | 2.58 | 0.010 | ** | 0.010* |
| Compensation receiver: base (male head) | | | | | | |
| Female head | 0.924 | 0.277 | 3.34 | 0.001 | *** | 0.205* |
| Both | −0.307 | 0.582 | −0.53 | 0.598 | | −0.074* |
| Compensation paid | 0.512 | 0.251 | 2.04 | 0.042 | ** | 0.063* |
| Extension service | 0.040 | 0.224 | 0.18 | 0.856 | | 0.025* |
| Off-farm employment | 0.762 | 0.243 | 3.13 | 0.002 | *** | 0.090* |
| Market access | 0.212 | 0.241 | 0.88 | 0.378 | | 0.050* |
| Training access | 0.723 | 0.244 | 2.96 | 0.003 | *** | 0.024* |
| Social network | 0.528 | 0.262 | 2.01 | 0.044 | ** | 0.033* |
| Credit access | −0.051 | 0.246 | −0.21 | 0.836 | | −0.011* |
| TLU | 0.762 | 0.298 | 2.56 | 0.011 | ** | 0.011 |
| Constant | −1.292 | 1.114 | −1.16 | 0.246 | | |
| Pseudo r-squared | 0.3661 | | Number of obs | 372 | | |
| LR Chi-square (22) | 147.59 | | Prob > chi2 | 0.000 | | |
| Log likelihood | −127.80179 | | | | | |

*** p<.01, ** p<.05, * p<.1.

(*) dy/dx is for discrete change of dummy variable from 0 to 1.

Source: own estimation, 2024.

human capital, providing them with better knowledge, skills, and the ability to process information. This finding resonates with broader literature on livelihood resilience, which emphasizes that better-educated household heads are better equipped to navigate post-displacement challenges [30,34]. This enables them to better understand and pursue a variety of livelihood options. Additionally, education can open up new opportunities for employment, entrepreneurship, and access to various support services and markets, while also improving adaptability, risk management, and decision-making abilities. These findings highlight the crucial role of human capital in enhancing the adaptive capacity of peri-urban farming communities facing land dispossession. The marginal effect reveals that, all else being equal, the probability of a household diversifying its livelihoods increases by 13.4% for those with higher levels of education. This result aligns with studies by [25,26,29], all of which emphasize the positive role of education in facilitating livelihood diversification among households affected by land expropriation and other shocks. This insight is aligned with findings in countries like China, where despite high levels of education, landless farmers struggle to transition into new livelihoods without adequate support from local governments [14].

 

The analysis also shows that marital status significantly influences the likelihood of livelihood diversification after compensation. Married households are more likely to diversify their livelihoods than single-headed households, as married couples typically share responsibilities and decision-making, which enhances their ability to adopt new livelihood strategies. The marginal effect indicates that married households have a 10.4% higher probability of diversifying their livelihoods compared to single, divorced, or widowed households. This finding emphasizes the role of social cohesion in enhancing adaptive capacity, aligning with international studies on family structures and resilience in the face of land expropriation [47]. This finding underscores the importance of marital status in shaping the adaptive capacity of households facing land dispossession, aligning with the work of [29]. However, it also raises important questions about gender equity and intra-household decision-making processes. While married households are better positioned, there is a growing need to explore gendered differences in how compensation is allocated and utilized within households, as female-headed households may face additional challenges in accessing resources and markets [33].

Household size, on the other hand, has a negative and statistically significant effect on livelihood diversification at the 10% significance level. Larger households tend to use compensation funds primarily for daily consumption and meeting the basic needs of the household, limiting their ability to invest in alternative livelihood strategies. In contrast, smaller households may have more resources and flexibility to explore new income-generating activities. The marginal effect reveals that for each additional household member, the probability of diversification decreases by 0.6%. This inverse relationship between household size and livelihood diversification is consistent with findings from [25,29] which highlight the challenges larger households face in redirecting compensation resources toward livelihood diversification. Moreover, this is consistent with global studies showing that larger households often face greater challenges in adopting new strategies due to the higher number of dependents and a greater reliance on subsistence farming [14].

Gender also plays a significant role in livelihood diversification after compensation. Female-headed households are more likely to diversify their livelihoods than male-headed or joint-headed households, with the likelihood increasing by 20.5% when the household head is female. This suggests that female-headed households may allocate compensation resources more toward diversifying income sources rather than focusing solely on immediate consumption needs. This finding is in line with [26], who also observed a greater inclination for female-headed households to diversify their livelihoods. Besides, this is in line with studies that have found women in similar contexts often pursue livelihood diversification strategies as a means of mitigating the economic impacts of land loss. The higher likelihood of diversification among female-headed households underscores the importance of gender-sensitive policies that not only ensure fair compensation but also provide targeted support for female entrepreneurs in peri-urban settings [47]. The size of the expropriated land is another important factor influencing livelihood diversification. The results indicate that households with larger expropriated land sizes are less likely to diversify their livelihoods after receiving compensation. The marginal effect analysis shows that for each additional unit increase in the size of the expropriated land, the probability of livelihood diversification decreases by 10.8%. While larger landholdings result in higher compensation, these households may be more inclined to reinvest the compensation into their existing agricultural activities or use it for immediate consumption rather than exploring alternative livelihoods, as found by [24]. While compensation payments are typically higher for those with larger landholdings, these households tend to reinvest compensation in their agricultural activities or use it for consumption, rather than exploring alternative livelihoods [34]. This raises important policy questions about the adequacy of compensation, particularly in contexts where land is not only an economic asset but also a social and cultural resource [47].

The perception of fair compensation also significantly affects the likelihood of livelihood diversification. When households perceive the compensation as fair and adequate, they are more likely to engage in livelihood diversification. The marginal effect analysis reveals that the probability of livelihood diversification increases by 6.3% for households that received fair compensation. This underscores the importance of fair compensation in fostering social justice and enabling households to explore alternative livelihood options, rather than relying solely on compensation for immediate needs. This finding aligns with studies from other developing countries, which highlight the link between perceived fairness in

compensation and increased social capital and community cohesion [47]. Fair compensation is not only an economic issue but also a critical element of social justice in the land expropriation process. In Ethiopia, where land tenure is often viewed as a key cultural asset, ensuring that compensation is seen as fair is crucial for fostering long-term resilience in peri-urban communities. These results align with [30]. The study also finds that participation in off-farm employment positively influences livelihood diversification. Households engaged in off-farm economic activities are more likely to diversify their livelihoods, with the marginal effect showing a 9.0% higher likelihood compared to households that do not participate in off-farm work. This is consistent with [46], who highlighted the role of off-farm employment in promoting livelihood diversification among households affected by land-related changes. These findings are consistent with the broader resilience literature, which highlights the importance of diversified income sources and access to social capital in mitigating the impacts of land expropriation [14]. However, farming as a primary occupation has a negative and statistically significant effect on livelihood diversification, with the probability of diversification decreasing by 3.5% for households that rely on farming as their main occupation. This suggests that farming may limit households' capacity or willingness to adopt new livelihood strategies after compensation, possibly due to a lack of skills, resources, or mindset to transition away from farming. This finding aligns with the work of [15].

Access to alternative land also has a positive and statistically significant impact on livelihood diversification. The marginal effect indicates that when households have access to alternative land, the probability of diversification increases by 1%. Access to alternative land provides additional resources and opportunities, facilitating the adoption of new livelihood activities, as noted by [24]. Similarly, access to training services has a positive and significant effect on livelihood diversification, with the marginal effect showing a 2.4% increase in the likelihood of diversification when households have access to training. Training helps households acquire new skills, knowledge, and capabilities, enabling them to adapt to the disruption caused by land expropriation and pursue alternative livelihood options. Furthermore, the importance of training and skill-building programs cannot be overstated. As observed in other developing countries, access to training helps farmers develop alternative skills, thus enhancing their ability to engage in non-agricultural livelihoods and reducing their vulnerability to economic shocks [35]. This finding is consistent with [1], who also emphasized the role of training in supporting livelihood diversification.

Social networking also plays a significant role in livelihood diversification. Households that maintain strong social networks are more likely to diversify their livelihoods, with the marginal effect showing a 3.3% higher likelihood of diversification. Social networks provide valuable resources, information, and support, facilitating access to new opportunities, markets, and credit. These findings align with [1], who highlighted the importance of trust-based networks in supporting livelihood diversification. Finally, livestock holdings (measured in Tropical Livestock Units, TLU) positively influence livelihood diversification, with a 1.1% higher probability of diversification for households with larger livestock assets. Livestock can serve as both capital and a buffer against shocks, providing a source of wealth that can be leveraged to pursue new livelihood strategies. Larger livestock holdings provide households with a buffer against shocks and serve as a source of capital for reinvestment. This finding underscores the need for policies that support asset accumulation in peri-urban farming communities, particularly in regions where land expropriation threatens traditional livelihoods. This finding is consistent with [30] and [48], who also noted the role of livestock as a valuable asset for livelihood diversification.

## 5. Conclusion and policy implications

In conclusion, urban expansion in developing countries like Ethiopia has become a pressing issue, especially for peri-urban farmers who face land expropriation and inadequate compensation. In these areas, the conversion of agricultural land to urban land use is a common practice, often driven by government authority, with farmland being seized without the consent of landholders. Ethiopian cities are rapidly expanding, encroaching on peri-urban poor communities' land to meet the growing demands for housing and infrastructure. This study examined the impact of urban expansion and land expropriation on the livelihoods of peri-urban farming households in the cities of North Wollo Zone, Ethiopia. This study

employed mixed research approach and multistage sampling method was used to select 378 expropriated household heads from four cities of the zone, namely, Woldia, Kobo, Mersa and Lalibela cities. This study employed both descriptive and econometric methods of data analysis.

The results show that after expropriation, households have sought to diversify their livelihoods through various economic activities, including agriculture, services, trade, manufacturing, construction, and mining. Despite these efforts, they continue to face significant challenges such as high inflation, insufficient compensation, lack of awareness, inadequate follow-up support, family disintegration, skill gaps, and informal land sales, all of which hinder their recovery and adaptation. Additionally, barriers like limited access to extension services, legal support, and the misallocation of compensation funds toward immediate consumption instead of long-term investment have further prevented households from building sustainable livelihoods. Furthermore, the logit estimated results reveal that, a household's livelihood diversification decision after land expropriation is mainly affected by household characteristics, socioeconomic characteristics and, institutional factors. Specifically, the livelihood diversification decision is positively affected by the education level of the head, marital status of the head, access to alternative land, and social networks; and negatively affected by household size, farming occupation, and the size of expropriated land. Finally, unfair compensation and the shortage of available land through formal channels have led to an increase in informal land transactions, driven by fears of unjust expropriation. The primary issue lies not just in the inadequacy of compensation but also in the misuse of even the small amounts paid to affected farmers. Therefore, it is crucial to support farmers in using the compensation funds for alternative businesses, and this support should be provided before expropriation occurs. A clear policy is needed that enables peri-urban farmers to adapt to urban life and integrate into urban society prior to land expropriation.

To improve compensation practices and support sustainable livelihoods for affected peri-urban farmers, this study recommends a multi-tiered and evidence-based approach. At the local level, government and municipal offices must actively engage with affected communities through consultations to co-develop fair compensation guidelines before implementation. Raising awareness, offering capacity-building training, and providing technical and legal support are essential to help farmers navigate the transition and effectively utilize compensation for long-term productivity. Urban planning should prioritize protecting fertile farmland by excluding it from development zones, while also promoting intensive farming and non-farm employment, especially for youth in expropriated households. Vertical housing development should be considered to optimize land use and preserve agricultural areas. Strengthening local social networks, such as farmer cooperatives, can enhance knowledge-sharing and collective bargaining for better compensation outcomes. At the regional level, authorities should monitor the implementation of compensation mechanisms and invest in programs that create alternative livelihood opportunities through micro-enterprises and access to credit, particularly for women and youth. Nationally, robust policy reforms are needed to establish transparent, fair land expropriation practices, such as updated asset valuation methods and land-to-land compensation, while securing land tenure to reduce informal transactions and foster sustainable investment. Finally, ensuring compensation reflects inflation and local conditions, providing alternative land, and facilitating employment integration will be critical for helping expropriated households build resilient and diversified livelihoods.

### 5.1. Future research area

Due to data limitations, secondary data, particularly on the legal frameworks and institutional settings, were not incorporated. Future research should address the legal and institutional aspects of land expropriation by incorporating secondary data on the legal framework and the role of local authorities. Additionally, this study employed a cross-sectional design, which provides a snapshot of the situation at a single point in time. Therefore, a longitudinal study would be valuable to track the long-term effects of land expropriation on livelihoods and assess the sustainability of diversification efforts, providing deeper understandings into the evolving challenges and effectiveness of policies over time.

## Author contributions

**Conceptualization:** Mesele Belay Zegeye, Moges Asmare Sisay, Dagnaw Beza Ayalew.

**Data curation:** Mesele Belay Zegeye, Dagnaw Beza Ayalew.

**Formal analysis:** Mesele Belay Zegeye, Moges Asmare Sisay, Dagnaw Beza Ayalew.

**Funding acquisition:** Mesele Belay Zegeye, Moges Asmare Sisay.

**Methodology:** Mesele Belay Zegeye, Dagnaw Beza Ayalew.

**Software:** Mesele Belay Zegeye.

**Supervision:** Mesele Belay Zegeye.

**Writing – original draft:** Mesele Belay Zegeye, Moges Asmare Sisay.

**Writing – review & editing:** Mesele Belay Zegeye, Moges Asmare Sisay, Dagnaw Beza Ayalew.

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
