## [Decision Letter · Decision Letter 0]

Dear Dr. Zegeye,

Thank you for submitting your manuscript to PLOS ONE. After careful consideration, we feel that it has merit but does not fully meet PLOS ONE’s publication criteria as it currently stands. Therefore, we invite you to submit a revised version of the manuscript that addresses the points raised during the review process.

We look forward to receiving your revised manuscript.

Kind regards,

Carolyn Chisadza

Academic Editor

PLOS ONE

Journal Requirements:

“The author(s) received specific funding for this work from Woldia University, Ethiopia.”

Please state what role the funders took in the study.  If the funders had no role, please state: 'The funders had no role in study design, data collection and analysis, decision to publish, or preparation of the manuscript.'

Additional Editor Comments:

The focus of the paper is interesting and topical. However, the authors need to make substantial improvements to it.

1. Clarify the methodology strategy. It is not clear where the secondary data is incorporated in the analysis as most of the findings appear to come from the survey undertaken.

2. Detail the variables that entail secondary data from the institutional reports related to land expropriation and compensation. Which institutional reports are these, what year and source of reports.

3. Pay close attention to the comments made by both reviewers as they overlap especially in explaining methodology criteria clearly, interpreting findings clearly with policy implications, and providing context in the literature review.

Reviewers' comments:

Reviewer's Responses to Questions

**Comments to the Author**

1. Is the manuscript technically sound, and do the data support the conclusions?

Reviewer #1: Partly

Reviewer #2: Yes

2. Has the statistical analysis been performed appropriately and rigorously?

Reviewer #1: No

Reviewer #2: Yes

3. Have the authors made all data underlying the findings in their manuscript fully available?

Reviewer #1: Yes

Reviewer #2: Yes

4. Is the manuscript presented in an intelligible fashion and written in standard English?

Reviewer #1: Yes

Reviewer #2: Yes

Reviewer #1: 1. Summary of the Research

• Overall Purpose: This paper aims to investigate the impact of land expropriation and compensation on the livelihood diversification strategies of peri-urban farmers in Woldia city and selected towns within the North Wollo Zone of Ethiopia. The authors employ a mixed-methods approach, incorporating both quantitative and qualitative data analysis to explore this complex issue. They analyze survey data from 372 households to identify the factors that influence livelihood diversification strategies.

• Context: Land expropriation for urban development is a growing concern in developing countries, with significant implications for affected communities. This study addresses a critical need to understand the socio-economic consequences of such processes, and how people adapt in response.

• Main Claims and Conclusions: The study argues that while many households have diversified their livelihoods, several factors hinder their long-term economic resilience. These include: inadequate compensation, limited access to extension services, and a lack of targeted support programs.

o The quantitative analysis identifies education, marital status, access to alternative land, and social networks as significant predictors of livelihood diversification.

o The qualitative component reveals issues of procedural unfairness, a lack of public consultation, and a widespread lack of guidance in the use of compensation funds.

• Strengths: The study tackles a relevant and timely topic in a rapidly urbanizing context. It gathers significant empirical data (though the analysis needs refinement), and explores the complex interplay between policy and individual household responses to land expropriation. The references are relatively comprehensive.

• Weaknesses: The major weakness of the paper lies in its inadequate use of the mixed method approach. There's a disconnect between the stated methodology and the actual analysis. The study design is not clearly articulated, and the qualitative findings do not seem well integrated into the discussion. Additionally, the analysis itself needs substantial work to provide better insights into the results.

Overall Recommendation: This manuscript requires major revisions before it can be considered for publication. The current version has significant methodological and analytical shortcomings.

2. Examples and Evidence

Major Issues

1. Title Inadequacy & Lack of Insights: The title promises "insights," but the manuscript mostly presents results with limited analysis or in-depth discussion. The findings should be linked to established theory and policy implications, specifically the insights that the title promises. The title needs to reflect the limitations of the work as well (e.g., "Livelihood Diversification Strategies") and may benefit from being more specific (e.g., "A Mixed-Methods Analysis").

2. Mixed Methods Inconsistency: The manuscript claims a mixed methods approach, but the discussion heavily focuses on quantitative findings. The qualitative data and their themes are not explicitly presented. This makes the approach feel more quantitative and misses key opportunities for deeper insights, which requires a major re-working of the manuscript.

3. Unclear Justification for Mixed Methods: The introduction mentions "mixed methods" but does not explain why this approach was chosen, and why that is essential. The authors must explain how the qual/quant approaches complement each other. (For example, do qualitative data refine the quant results?) This must be done in the methods and introduced in the research design section.

4. Missing Qualitative Analysis: The method mentions thematic analysis, yet there is no explicit discussion of the themes or how they were identified. Quotes or examples from interviews and focus groups should be included to ground the discussion. This is a major shortcoming that must be corrected.

5. Research Design Vague: The paper lacks a clear description of the study design. The manuscript needs to articulate:

o The specific mixed methods design (e.g., convergent parallel, explanatory sequential).

o How the quantitative and qualitative data were integrated (or not, in this case).

6. Problem Statement and Gap: The problem statement should clearly explain the existing research and knowledge gap that this study fills. The gap is not clear in the manuscript.

7. Statistical Rigor: There is minimal discussion of the statistical methods beyond mentioning "binary logistic regression." There is no explanation as to why the model variables were chosen, or if any model specification tests were run.

8. Affiliation Format: As noted, affiliations are overly detailed. Simply state the department or faculty, and the institution, for example, Department of Economics, Woldia University.

9. Abstract Weakness: The abstract lacks specificity, the gap is not clear, and does not summarize the key findings. This needs to be revised, making sure that the methods used are clearly stated and the summary points of the results are included.

Minor Issues

1. Study Area Maps: Including a map to visualize the study area is important and a suggestion that the authors should strongly consider. This will help readers understand the geographical context.

2. Visual Flow: Adding a flow chart or diagram that describes the methods would help readers understand the study.

3. Overuse of "Peri-Urban": The term "peri-urban" is frequently used. It is ok, but should be better described as to how this study area fits into the definition of a peri-urban environment, and if other studies define the study area as such.

4. Table Clarity: Tables could be improved (i.e., table 2), which need better headings and subheadings for better readability and interpretation. Tables should have a clear and concise explanation or interpretation of the results in the text.

5. Variable Interpretation: the variables in table 1 need to be described a bit better. The reader cannot intuit all the meanings of all the variables without explanation.

3. Other Points

• Ethical Concerns: The paper mentions ethical approval and informed consent but should include more detail (e.g., if participants were told that participation was voluntary or that their data could be shared anonymously).

• Availability for Review: Yes, I am available to review a revised version of this manuscript.

In summary, the authors need to substantially revise the manuscript. The major deficiencies must be corrected before the work can be recommended for publication.

I hope this critique is helpful. Please let me know if you have any further questions.

Reviewer #2: Land Expropriation, Compensation, and Livelihood Diversification: Insights from Peri-

Urban Farmers in Woldia city and selected Towns of North Wollo Zone, Ethiopia

Comments

I enjoyed reading this paper. The authors can consider the following comments to improve their work:

• Consider providing a more concise historical context of urbanization's impact specifically in the study area to ground the research in a specific socio-economic context.

• Consider adding an area of study map

• Clarify and elaborate on the unique characteristics of peri-urban farming in Woldia and selected towns compared to other regions in Ethiopia, as this can strengthen the argument about the significance of studying this specific area.

• I find the paper lacking a robust literature review, conceptual or theoretical section (this could focus on 1. global and regional trends in urbanization, especially in developing countries, highlighting how urban expansion affects peri-urban lands and communities, 2. theories related to livelihood diversification and compensation, such as the Sustainable Livelihood Framework, to understand the strategies households employ in response to land loss, 3. Review empirical studies that document the social, economic, and psychological effects of land expropriation on peri-urban farmers, focusing on both challenges and adaptive strategies. 4. Analyzing the specific legal and policy frameworks governing land expropriation in Ethiopia, noting recent changes and their implications for land compensation and livelihood restoration. 5. Identifying gaps in existing literature, particularly concerning long-term impacts on livelihoods and the effectiveness of compensation measures, justifying the need for the current study’s focus.

• The sample size and sampling technique are well-explained, but the paper could benefit from a more detailed discussion on the limitations of the chosen methods, especially concerning potential biases in key informant interviews and focus group discussions.

• Consider discussing the validation of the structured questionnaires used for collecting quantitative data to enhance credibility.

• While the use of binary logistic regression is appropriate, ensure the model assumptions and the choice of variables are justified in more detail. This can include discussing why certain variables were included or excluded and the potential impact of this on the findings.

• Provide a more critical analysis of the quantitative data. For instance, discussing the implications of high inflation and lack of urban living skills on livelihood diversification could add depth.

• Results are comprehensive but could be better organized to lead the reader through the findings more clearly. Consider using subheadings to distinguish between different aspects of the results, such as economic impacts versus social impacts.

• A separate strong discussion might be good for this paper (Discuss how the findings align with or differ from theoretical expectations related to livelihood diversification and compensation, Compare the results with similar studies in other regions or contexts, highlighting unique findings and common trends., Based on the findings, recommend specific policy changes or interventions that could improve compensation practices and support livelihood diversification for affected farmers., Discuss practical measures that local governments and development agencies can implement to mitigate the negative impacts of land expropriation on peri-urban farmers.

• The policy implications are practical; however, they could be strengthened by suggesting specific policy measures or programs, detailing how they could be implemented within the current Ethiopian administrative and legal framework.

• Consider adding future research directions, suggesting specific areas that need further investigation based on the study's findings.

• Proof read the paper

**Do you want your identity to be public for this peer review?** For information about this choice, including consent withdrawal, please see our Privacy Policy

Reviewer #1: No

Reviewer #2: **Yes: ** Dr Johannes Bhanye

---

## [Decision Letter · Decision Letter 1]

Dear Dr. Zegeye,

We look forward to receiving your revised manuscript.

Kind regards,

Carolyn Chisadza

Academic Editor

PLOS ONE

Additional Editor Comments:

Please pay close attention and address the comments by the reviewer.

Reviewers' comments:

Reviewer's Responses to Questions

**Comments to the Author**

Reviewer #1: (No Response)

Reviewer #2: All comments have been addressed

2. Is the manuscript technically sound, and do the data support the conclusions?

Reviewer #1: Partly

Reviewer #2: Yes

3. Has the statistical analysis been performed appropriately and rigorously?

Reviewer #1: Yes

Reviewer #2: Yes

4. Have the authors made all data underlying the findings in their manuscript fully available?

Reviewer #1: Yes

Reviewer #2: Yes

5. Is the manuscript presented in an intelligible fashion and written in standard English?

Reviewer #1: No

Reviewer #2: Yes

Reviewer #1: General Evaluation

Despite some improvements, the manuscript still lacks sufficient theoretical grounding, conceptual clarity, and integration with broader urbanization and livelihood transformation literature. The study remains largely descriptive, with insufficient analytical depth in interpreting empirical findings.

Key Issues

1. Misuse of the "Mixed-Methods" Label

The manuscript claims to use a mixed-methods approach, yet:

• The qualitative component is minimal and superficially integrated.

• There is no detailed thematic presentation or deep narrative analysis from interviews or FGDs.

• The qualitative findings are only used to validate quantitative results, not to enrich or challenge them—making the “mixed-methods” claim questionable.

Recommendation: Either significantly strengthen the qualitative section (e.g., coding themes, integrating narratives) or revise the paper to clearly present it as a quantitative-dominant study with supportive qualitative insights.

2. Research design:

• The actual execution of the qualitative component in the manuscript does not meet the expectations of an explanatory sequential mixed-methods design:

o Qualitative data are minimal (just a few quotes or summary comments from FGDs and KIIs).

o There is no structured qualitative analysis (e.g., coding themes, presenting participant narratives, triangulating).

o The qualitative findings are not used to meaningfully explain or reinterpret the quantitative findings.

3. Lack of a Clearly Articulated Research Gap

The introduction claims that the study “addresses several gaps in previous research” but fails to clearly define or articulate these gaps. It mostly reiterates findings from earlier studies and then claims that North Wollo Zone is understudied.

• No concrete theoretical or conceptual contribution is identified.

• The "gap" is geographical rather than scientific or methodological.

• The literature review is descriptive but not analytical or critical.

Recommendation:

• Clearly identify what is missing in existing research (e.g., lack of livelihood transition models, weak compensation policy impact analysis).

• Explain how this study adds new conceptual or empirical knowledge rather than just replicating existing frameworks in a new location.

4. Weak Conceptual Framework

• While a figure is presented, it is poorly explained in the text. The conceptual framework lacks connection to widely accepted livelihood frameworks such as the Sustainable Livelihoods Framework (SLF).

• There is no clear articulation of how urbanization, expropriation, and compensation relate causally to specific livelihood outcomes beyond descriptive mentions.

Recommendation: Embed the study within a recognized theoretical model (e.g., Ellis 2000, DFID 1999 SLF) and elaborate the conceptual linkages in text.

5. Limited Literature Integration

• The manuscript reads like a literature summary rather than a critical review. Many statements cite multiple sources but lack synthesis or critical engagement.

• International comparisons are limited; much of the cited literature is either Ethiopian or generic, missing insights from broader urban displacement and resettlement research.

Recommendation: Engage with recent, comparative urban land studies to better situate findings.

6. Poorly Constructed Study Area Maps

The maps are substandard. This remains true:

• The map of Ethiopia does not label all regions.

• The Amhara map lacks administrative zones.

• The North Wollo map does not clearly mark the selected towns or woredas.

Recommendation:

• Include three clearly labeled maps:

1. Ethiopia (showing all regions).

2. Amhara region (with all administrative zones).

3. North Wollo Zone (clearly highlighting study towns like Woldia, Kobo, Lalibela, Mersa).

• Use proper map scale, legends, north arrows, and standard cartographic conventions.

7. Underdeveloped Discussion Section

• The discussion lacks comparative insight and fails to clearly link results to research questions or broader debates on livelihood resilience or governance.

• Assertions about informality, policy failure, and livelihood decline are not substantiated with in-depth policy or institutional analysis.

Recommendation: Expand discussion by linking results to key debates—e.g., expropriation justice, participatory urban governance, resilience of peri-urban communities.

8. Conclusion and Policy Implications Are Vague

• The conclusion restates findings without advancing clear, actionable policy recommendations.

• It does not reflect on institutional responsibilities, nor does it address long-term adaptation strategies or urban policy reforms.

Recommendation: Provide specific, evidence-based policy recommendations, ideally tiered (e.g., local, regional, national) and connected to institutional gaps found in the study.

Reviewer #2: The authors significantly improved the paper. I comment the effort they made. I recommend publication of the paper

**Do you want your identity to be public for this peer review?** For information about this choice, including consent withdrawal, please see our Privacy Policy

Reviewer #1: No

Reviewer #2: **Yes: ** Johannes Bhanye

---

## [Editor Report · Decision Letter 2]

Effects of Urbanization and Land Expropriation on the Livelihoods of Peri-Urban Farmers in North Wollo Zone, Ethiopia

PONE-D-25-01172R2

Dear Dr. Zegeye,

We’re pleased to inform you that your manuscript has been judged scientifically suitable for publication and will be formally accepted for publication once it meets all outstanding technical requirements.

Kind regards,

Carolyn Chisadza

Academic Editor

PLOS ONE
---

## [Editor Report · Acceptance letter]

PONE-D-25-01172R2

PLOS ONE

Dear Dr. Zegeye,

I'm pleased to inform you that your manuscript has been deemed suitable for publication in PLOS ONE. Congratulations! Your manuscript is now being handed over to our production team.

Kind regards,

on behalf of

Prof Carolyn Chisadza

Academic Editor

PLOS ONE